# The Effects of Mono- and Bivalent Linear Alkyl Interlayer Spacers on the Photobehavior of Mn(II)-Based Perovskites

**DOI:** 10.3390/ijms24043280

**Published:** 2023-02-07

**Authors:** Soumyadipta Rakshit, Alicia Maldonado Medina, Luis Lezama, Boiko Cohen, Abderrazzak Douhal

**Affiliations:** 1Departamento de Química Física, Facultad de Ciencias Ambientales y Bioquímica and INAMOL, Universidad de Castilla-La Mancha, 45071 Toledo, Spain; 2Departamento de Química Orgánica e Inorgánica, Facultad de Ciencia y Tecnología, Universidad del País Vasco, UPV/EHU, Bº Sarriena s/n, 48940 Leioa, Spain

**Keywords:** photoluminescence, Mn (II), perovskite, octahedron, tetrahedron, LED, quantum yield, lifetime, electron-phonon coupling, Mn-Mn interactions

## Abstract

Mn(II)-based perovskite materials are being intensively explored for lighting applications; understanding the role of ligands regarding their photobehavior is fundamental for their development. Herein, we report on two Mn (II) bromide perovskites using monovalent (perovskite 1, P1) and bivalent (perovskite 2, P2) alkyl interlayer spacers. The perovskites were characterized with powder X-ray diffraction (PXRD), electron spin paramagnetic resonance (EPR), steady-state, and time-resolved emission spectroscopy. The EPR experiments suggest octahedral coordination in P1 and tetrahedral coordination for P2, while the PXRD results demonstrate the presence of a hydrated phase in P2 when exposed to ambient conditions. P1 exhibits an orange-red emission, while P2 shows a green photoluminescence, as a result of the different types of coordination of Mn(II) ions. Furthermore, the P2 photoluminescence quantum yield (26%) is significantly higher than that of P1 (3.6 %), which we explain in terms of different electron-phonon couplings and Mn-Mn interactions. The encapsulation of both perovskites into a PMMA film largely increases their stability against moisture, being more than 1000 h for P2. Upon increasing the temperature, the emission intensity of both perovskites decreases without a significant shift in the emission spectrum, which is explained in terms of an increase in the electron-phonon interactions. The photoluminescence decays fit two components in the microsecond regime—the shortest lifetime for hydrated phases and the longest one for non-hydrated phases. Our findings provide insights into the effects of linear mono- and bivalent organic interlayer spacer cations on the photophysics of these kinds of Mn (II)-based perovskites. The results will help in better designs of Mn(II)-perovskites, to increase their lighting performance.

## 1. Introduction

Researchers have known about perovskites (PSs) since the late 19th century; however, it was only after the pioneering work by Miyasaka and co-workers [1] that organic-inorganic perovskites received extraordinary attention, owing to their tremendous success in photovoltaic and related optoelectronic devices. One of the interesting features of these structures is their ability to accommodate large cations, which paves the way for small organic cations to participate in their framework, leading to organic-inorganic hybridization [2,3]. During the last five years, hybrid organic-inorganic PS-based solar cells and LEDs have experienced remarkable and unprecedented achievements [4,5,6,7,8,9].

Despite reaching important milestones in a short time, lead (Pb)-based perovskites suffer from unsatisfactory long-term stability due to mobile ionic features, and toxicity, which in turn reduce the possibility of their commercialization. Therefore, it is desirable to develop Pb-free stable and highly luminescent organic-inorganic hybrid perovskites with larger stability and excellent performance. To this end, a series of environmentally friendly alternatives, such as tin (Sn^2+^), copper (Cu^2+^), bismuth (Bi^3+^), and antimony (Sb^3+^) cations have been explored to replace the toxic Pb^2+^. Among these alternatives, Sn-based perovskites showed great potential, but unfortunately, the rapid oxidations of Sn^2+^ and Sn^4+^ drastically hinder their stability and, hence, their applicability [10,11,12]. However, for 2D Sn-based Ruddlesden-Popper perovskites, the sizes and configurations of organic spacer cations have remarkable effects on the properties of these materials [13,14].

Many researchers have turned their attention to Mn (II)-based organic-inorganic hybrid perovskites as alternative light emitters owing to their nonlinear optical properties and tunable luminescence [15,16,17,18,19]. Multiferroics and phase-change memories are also observed in these materials [20,21,22,23,24], as well as the coexistence of multiple performances. The emissions of these perovskites depend on the coordination environment of Mn^2+^ resulting from the ^4^T_1_-^6^A_1_ transition [15,25,26]. It has been established that octahedrally coordinated Mn^2+^ cations show orange or red emissions, whereas tetrahedrally coordinated ones exhibit green photoluminescence. Green emissions are instigated by two major factors: (i) the absence of an inversion center in the tetrahedral environment and (ii) increased electric-dipole oscillator strength due to the small crystal field splitting energy of [MnX_4_]^2−^. In the octahedral environment, the red emission is associated with a higher Mn-Mn coupling interaction resulting from short Mn-Mn distances (3–5 Å) [27,28]. Apart from the single green or red emission bands, dual emission has also been reported in a few cases due to the coexistence of both environments and/or from electron-phonon coupling variation [29,30,31]. Triboluminescence is another interesting phenomenon that has been observed in non-centrosymmetric crystals of Mn (II) complexes owing to its great potential in structural damage sensing, stress sensing, display, and security marking [32,33,34,35].

Substitution of halide ions and changing the organic counterions are also effective in modulating the optical and physical properties of these materials. For example, (pyrrolidinium)MnCl_3_ and (pyridine)MnCl_3_ octahedral coordination were reported to exhibit red/orange emissions while their bromide counterparts (pyrrolidinium)_2_MnBr_4_ and (pyridine)_2_MnBr_4_ were found to emit in the green region due to their tetrahedral coordination [15,36]. Two perovskites, (C_5_H_6_N)_2_MnBr_4_ and (C_5_H_6_N)MnBr_3_—with different crystal structures—have been synthesized by adjusting the amount of the pyridinium cation. The first one has an isolated mononuclear structure with a tetrahedron [MnBr_4_]^2−^ unit, and the second one has a linear chain with an octahedral [MnBr_6_]^4−^ part to give green and red emission bands, respectively [17]. Due to their crystal structure-dependent tunable optical properties and high emission quantum yields, Mn-based perovskites are now being largely considered in the development of new LED devices and as efficient X-ray scintillators [37,38]. However, there are still several issues to resolve before these materials are applied. In this regard, the destabilization of Mn-based perovskites in moisture is worth mentioning, as they are based on salts. In the presence of water, the ionic coordination bonds of Mn (II) centers can easily be destroyed or the material can undergo a phase transformation process, leading to a loss of the emission [39]. By heating or exposing these structures to aprotic solvent vapors, the photoluminescence can be partially or almost totally recovered. These characteristics have been exploited for the fabrication of rewritable PL papers [39] and in the sensing of acetone [40]. Another issue to consider for a better understanding of Mn-based perovskite spectroscopy is to explore the role played by the organic ligand on the structure and photobehavior. Few reports have suggested the involvement of electron-phonon interactions in their photochemistry [29,30,31]. To gain insight into the electronic structure, the origin of bright photoluminescence, and ferromagnetic coupling in these kinds of materials, several ab initio models have been proposed [41,42]. Recently, these methods have been applied to predict the band structures and validate the origin of Mn^2+^ photoluminescence through spin forbidden ^4^T_1_ → ^6^A_1_ transitions for some all-Mn halide perovskite single crystals [16,17,38,40].

In this work, we synthesized two Mn (II)-based perovskites using a monovalent (ethyl ammonium (EA) bromide) and a bivalent (ethyl diammonium (EDA) dibromide) interlayer spacer cation. The resulting perovskites, C_2_H_5_NH_3_MnBr_3_ (P1) and C_2_H_4_(NH_3_)_2_MnBr_4_ (P2), respectively, were characterized by powder X-ray diffraction technique (PXRD,) electron-spin paramagnetic resonance (EPR), steady-state, and time-resolved emission spectroscopy. We observed that the P1 perovskite, having monovalent cations, showed orange-red emissions, while P2, based on bivalent interlayer spacers, exhibits green photoluminescence. This difference results from a distinct Mn-coordination (octahedral in P1 and tetrahedral in P2). Furthermore, we found that P2 showed a higher photoluminescence quantum yield (26%) compared to 3.6% of P1. The photoluminescence lifetimes for P2 are 0.10 and 0.37 ms, assigned to hydrated and non-hydrated phases in the perovskites, respectively. The emission intensities of both perovskites exhibit large sensitivity to moisture (air humidity, 50–60%) and significantly improved stability upon encapsulation within a poly(methyl methacrylate) (PMMA) film. The temperature effect on the emission intensity of P2 suggests an activation energy barrier, Δ*E_a_* = 4.46 kJ/mol (Δ*E_a_* = 1.67 kJ/mol for P1) for the non-radiative processes, most probably due to electron-phonon interactions. To the best of our knowledge, this is the first report comparing the effect of mono- and bivalent spacer cations bounded to the Mn centers in Mn-based perovskites, using the same carbon chain. We believe that our results will help when designing new Mn(II) perovskites for further improvement of their photonic performances.

## 2. Results and Discussion

### 2.1. Structural Characterization

The synthesized perovskites (Figure 1) were characterized by powder X-ray diffraction (PXRD) and EPR techniques (Figure 1).

The PXRD patterns of dried P1 and P2, and hydrated P2 (short exposure, P2(H1) and 12-h exposure, P2(H2)) (Figure 1A) are comparable to those reported for similar Mn-based perovskites, reflecting their crystalline structures. Notably, the PXRD of P1 presented fewer peaks in comparison with those of P2, P2(H1), and P2(H2), which suggests a higher crystallinity of the former. Additionally, in the PXRD of P2, P2(H1), and P2(H2), we observed the presence of a peak at ~8.5°, which increased in intensity when the perovskite crystals were exposed to ambient conditions and allowed to adsorb water from the environment (P2(H1)). The PXRD pattern of P2(H2) exposed to the ambient humidity (50–60%) for a longer time (12 h) showed a notable increase in the intensity of the peak associated with the hydrated quasi-octahedral phase at 8.5°. Similar behavior with the appearance of a peak at around 10° of trans-2,5-dimethylpiperazine Mn(II) bromide perovskite was reported upon adsorption and coordination of water molecules [39]. In that perovskite, Mn^2+^ adopted a quasi-octahedral coordination sphere to produce a secondary, non-emissive phase along with the green emissive tetrahedrally-coordinated Mn^2+^ ions. We believe that upon hydration, a population of Mn(II) in P2 experiences a similar change in the Mn^2+^ ion configuration-giving structures, such as P2(H1) and P2(H2).

It should be noted that our attempts to obtain sufficiently large single crystals to perform X-ray experiments through slow solvent evaporation and antisolvent procedures were unsuccessful. The inability to obtain a macroscopic single crystal is most probably due to the flexibility of the linear aliphatic spacers that inhibit the growth of large crystals [43,44]. In a recent work, the impact of the organic A′ site ligand structure on the formation of MAPbI_3_ perovskite films was studied in terms of crystallization kinetics, precursor solutions, and crystal phase compositions [43]. The authors employed n-butylammonium (n-BA^+^) and iso-butylammonium (iso-BA^+^) ligands and found that changing from the linear n-BA^+^ to the branched iso-BA^+^ molecule led to better crystal orientation and improved film crystallinity. The observed effect was explained in terms of the spontaneous formation of large clusters (due to lower enthalpies of the accumulation of iso-BA^+^ versus n-BA^+^ ligand) in the precursor solution of iso-BA^+^, which could act as pre-nucleation sites to accelerate the crystallization of 2D perovskites [43]. Additionally, for P2, which is synthesized in an aqueous HBr medium, the precursor (protonated EDA) is highly soluble in water (and insoluble in almost any other solvent), which further impedes the formation of a single P2-crystal in this environment. Therefore, we could not obtain direct information about the structures of P1 and P2, the involved distances between the Mn clusters, and the types of interactions.

EPR is a suitable technique used to determine the oxidation state, spin state, and local coordination of paramagnetic ions. To this aim, X-band EPR spectra were registered on powdered samples of P1 and P2 at room temperature. An ideally resolved EPR spectrum of isolated Mn^2+^ ions consists of five signals corresponding to the fine structure (S = 5/2) that further splits into six lines due to the hyperfine interactions (^55^Mn, I = 5/2) [45]. However, in the studied samples, the structures appear to be collapsed (Figure 1B). The observed resonances are centered at *g* values very close to the free electron one, as expected for high-spin Mn^2+^ ions since orbital contributions to their magnetic moments are not expected in either octahedral or tetrahedral environments (^6^A_1g_ and ^6^A_1_ terms, respectively) [46]. The EPR spectrum of P1 can be well-fitted by a single Lorentzian line with g = 2.007 and the peak-to-peak linewidth, ΔH_pp_ = 12.1 mT. The absence of fine and/or hyperfine structures and the Lorentzian shape of the curve indicate an exchange narrowing of the spectrum due to strong magnetic interactions that average the local fields around the Mn^2+^ ions in this compound [47,48]. Similar spectra have been found in other hybrid perovskites with manganese in octahedral environments [49,50].

On the contrary, the EPR spectrum of the P2 sample shows a very broad and slightly anisotropic line (g = 2.01; ΔH_pp_ = 140 mT), together with a very weak and narrower signal that could correspond to a secondary phase. The broadened appearance of the P2 spectrum can be attributed to either dipole–dipole interactions, incomplete resolution of the fine structure, or shortening of the spin–lattice and/or spin–spin relaxation times. It is well known that dipolar interactions modify the local fields felt by the electron spins leading to an increase in the linewidth. On the other hand, distortions of the crystalline field remove the degeneracy of the free Mn^2+^ ground state ^6^S_5/2_ into three Kramer doublets (±5/2, ±3/2, and ±1/2) enabling five allowed EPR transitions for each orientation. In powder spectra, the averaging and overlapping of these signals often result in broad signals [51]. Moreover, when the distortion is high and leads to zero-field splitting of the same order as the Zeeman splitting, the EPR spectrum is usually broad because the spin system is strongly coupled to vibrational modes and. hence, the relaxation times become quite short [52,53]. Therefore, the large difference between line widths shown by the EPR spectra of P1 and P2 suggests that the environment of the Mn^2+^ ions is different in both cases. The zero-field splitting appears to be higher for the P2 compound, as expected for manganese (II) in a distorted tetrahedral environment [54]. Thus, although we do not have a structure based on single crystal X-ray experiments, from these EPR results, the color of the observed emission of P1 (orange/red) and P2 (green), and the published reports on Mn-based perovskites, we believe that Mn^2+^ ions in the former have octahedral coordination, while in the later it is a tetrahedral one (Figure 1). In these proposed structures, where an Mn-bromide cluster interacts with the ammonium group of protonated EA and EDA spacers, we invoke the formation of H-bonds to form and maintain the perovskite structure, as it has been reported in other reports. [15,38,55] Thus, in the following sections, we adopt the molecular structures of the complexes shown in Figure 1.

### 2.2. Photophysical Characterization

#### 2.2.1. Steady-State UV-Visible Spectroscopy

To gain insight into the photobehavior of these perovskites, we recorded solid-state UV–vis diffuse reflectance, excitation, and emission spectra at ambient conditions (Figure 2). The diffuse reflectance and the excitation spectra present several strong peaks at the UV and visible regions. They reflect an electronic transition from the ^6^A_1_(S) low-lying state of Mn^2+^ ions to the excited states of the MnBr_x_ clusters [28]. We observed no dependence of the excitation spectra with the emission wavelength.

First, we show and discuss the results for P1. The peaks in the diffuse reflectance and the excitation spectra are located at around 366, 377 (a shoulder), 427, 438 (a shoulder), and 524 nm. These arise from the ^6^A_1_(S) → ^4^E(D), ^6^A_1_(S)→ ^4^T_2_(D), ^6^A_1_(S) → (^4^A_1_, ^4^E(G)), ^6^A_1_(S) →^4^T_2_(G) and ^6^A_1_(S) → ^4^T_1_(G) transitions of the Mn^2+^ ions, respectively (Figure 2A). Most importantly, the peaks at 427 and 524 nm are typical features of octahedrally coordinated Mn^2+^ ions [28]. P1 displays a red-light emission upon UV–light irradiation and its emission spectrum consists of a single band located at 602 nm with a full width at half maximum (FWHM) intensity of 70 nm (1932 cm^−1^) that is independent of the excitation wavelength (Figure 2A). Based on previous reports of Mn-based perovskites and the observed orange/red emission of P1, we suggest octahedral coordination of Mn^2+^ in this phase, in agreement with the EPR results [17,28,39]. The measured photoluminescent quantum yield (PLQY) for P1 gives a value of 3.6%, which is lower than the one for P2 having two ammonia centers to interact with two [MnBr_4_]^2-^ clusters.

For P2, the intense peaks in the excitation spectrum collected at 545 nm are located at around 362, 371, 389, 435, 449, and 465 nm and according to the literature they arise from the ^6^A_1_(S)→ ^4^T_2_(P), ^6^A_1_(S) → ^4^E(D), ^6^A_1_(S)→ ^4^T_2_(D), ^6^A_1_(S) → (^4^A_1_, ^4^E(G)), ^6^A_1_(S) →^4^T_2_(G) and ^6^A_1_(S) → ^4^T_1_(G) transitions of the Mn^2+^ ions (such as in P1), respectively (Figure 2B). The positions of these peaks are consistent with the energy states splitting for Mn^2+^ in a tetrahedral environment [28]. Notably, the diffuse reflectance spectrum of P2 shows the presence of additional peaks at 425 and 510 nm that most probably arise from a secondary population of Mn^2+^ ions with quasi-octahedral coordination [39]. The presence of this secondary phase is further supported by the PXRD and EPR results, as discussed above, and by the diffuse reflectance spectrum of the hydrated P2 (P2(H1)), where the intensity of these peaks increases (dashed line, Figure 2B). However, contrary to the reported emissive behavior of Mn^2+^ with octahedral coordination in other Mn-based hybrid materials, [28,37,56] for P2 (and P2(H1)), the difference in the relative intensity of the peaks in the diffuse reflectance and excitation spectra indicates that these hydrated Mn(II) centers are non- or weakly-emissive. Recently, it was suggested that the emissive tetrahedral Mn^2+^ ions in trans-2,5-dimethylpiperazine manganese (II) bromide can be transformed to a non-emissive hydrated phase by adsorbing water molecules and the Mn^2+^ ions could adopt a quasi-octahedral coordination sphere [39]. Thus, the peaks observed in the diffuse reflectance spectra of P2 and P2(H1) at 425 and 510 nm arise from the non-emissive secondary hydrated phase. The origin of this phase could be due to the recapturing of water molecules from the ambient atmosphere (humidity of 50–60%). On the other hand, it could be the result of the synthesis procedure, where water is the used solvent (HBr, 48% aqueous solution) and it could coordinate with Mn^2+^ to produce the non-emissive population with octahedral coordination. Under UV–light irradiation (365 nm) P2 emits a strong green light stemming from the highly localized intra-atomic Mn^2+^ d-d transitions (inset of Figure 2B) [28]. The emission spectrum consists of a single band with the maximum intensity located at 544 nm and the FWHM at 53 nm (1742 cm^−1^), which is also independent of the excitation wavelength (Figure 2B). Next, we measured the PLQY at ambient conditions. The obtained PLQY value of 26% is large, but lower in comparison with other green-emissive Mn-based hybrid materials with tetrahedral coordination that have reported values >80% [15,25,26,36,39,40,57]. The most probable explanation for the observed discrepancy is the presence of the non-emissive hydrated phase with octahedral coordination of the Mn^2+^ ions in our perovskite.

To support this explanation, we recorded the emission intensity of the neat crystals of P1 and P2 exposed to air and of crystals covered by a PMMA film over long period of time (Figure 3). While the emission intensity of the neat crystals drastically drops with time (100% in 6 h for P1 and in 8 h for P2) due to the adsorption of water, upon protection with a PMMA film, the intensity initially drops by only 25% after 40 h for P2 and 20% after 25 h for P1. Following this initial drop, the emission intensity of the protected P2 remarkably remains constant for a long time (1000 h). The initial drop in the emission intensity for the PMMA-protected samples at the first hours is most probably due to interaction of the perovskite crystals with the used solvent (toluene) to prepare the PMMA film.

Now, we turn our attention to the differences in the steady-state absorption and emission spectra of P1 and P2 and compare them with those of other Mn-based perovskites. Several factors are expected to contribute to the photophysical behavior of the studied structures, some of which are more specific for Mn^2+^-perovskites, such as the Mn-Mn distances and electron-phonon couplings, while others are more general and can be extrapolated from other types of perovskites (such as H-bonding and other non-covalent interactions or structural arrangement [58,59,60]). Related to the structure, we believe that the observed difference in the behavior of P1 and P2 could stem from differences in the structural arrangement of the Mn-bromide clusters due to the used organic ligand. We hypothesize that the EA organic monocation in P1 most probably gives rise to either a 3D hybrid organic–inorganic perovskite (HOIP), BMX_3_ (B = small organic cation; X = halide and M = metal) or a structure similar to Ruddlesden–Popper-like 2D layered HOIP [60,61]. In both cases, these allow for shorter distances between the luminescent Mn centers resulting in a stronger coupling. On the other hand, the EDA dication in P2 is expected to give rise to Dion–Jacobson-like 2D HOIP layered structures. In these, we expect that the structures exhibit perfect stacking with no offset or displacement between successive layers, which can be attributed to the single layer of ions between each layer, as has been reported for other perovskites [61].

The structural organization of P1 and P2 is closely related to the coordination sphere of Mn^2+^. It is well established that the linear chain of the octahedron coordination, such as in the P1 sample, gives rise to a system of exchange-coupled Mn^2+^ ions since the nearest Mn-Mn distance is much shorter (3–5 Å) than that of Mn-Mn within the nearest neighboring chain [18,28,37,56]. This enables intrachain Mn-Mn coupling several orders of magnitude stronger than the interchain Mn-Mn coupling. Thus, the Mn^2+^ ions arrangement in these systems provides excitonic confinement giving rise to a substantial reduction of exciton capture by nonradiative traps. In addition to that, the interaction between the nearest Mn^2+^ ions within the linear chain is beneficial for enhancing Mn^2+^ absorption [28,37,56]. Furthermore, it was suggested that for systems with tetrahedral arrangements, such as P2, the Mn-containing [MnBr_4_]^2−^ tetrahedron is individually isolated among the crystal lattice and separated from the other tetrahedrons by the large organic cation (EDA in the case of P2) [25,37,62]. The longer distance between two neighboring [MnBr_4_]^2−^ tetrahedra, which is estimated between ~7 and 10 Å, can reduce the Mn-Mn interaction and hence significantly inhibit the concentration quenching effect of Mn^2+^ by suppressing the migration/dissipation of excitation energy to adjacent luminescent centers of Mn^2+^ [25,27]. As a result, it is possible that longer Mn-Mn distances would enable all Mn^2+^ luminescent centers to emit simultaneously under excitation irrespective of crystal defects thereby achieving a higher PLQY. Furthermore, in previous studies that have reported notably higher PLQY, the used organic cations rendered significantly more rigid structures [15,25,28,33,37,39,62]. This increased rigidity can reduce the thermal vibration of the inorganic anions, which in turn can suppress the non-radiative transitions and result in a significantly higher PLQY. In P1 and P2, the organic cations have linear chains that render higher flexibility in the resulting lattice, which would enhance the electron-photon interactions and might in part explain the relatively lower PLQY values.

Independently on whether the obtained structures are 3D or 2D HOIP, the strength of the non-covalent interactions (H-bonding, halogen bonding, van der Waals, to name a few) plays a significant role in the structural stability and phase transformation [58,59,60,63]. In these structures, the organic cations interact with the surrounding metal clusters through weak secondary H-bonding, with energies lower than 0.1 eV per bond [58,64]. DFT calculations on MAPbX_3_ (MA = methyl ammonium; X = Br or I) have revealed the role of the different H-bonding interactions in the stability of the perovskite network [65,66]. Due to these weak interaction energies, the reorientation of the organic cation can be activated either thermally at a finite temperature or by applying mechanical stimuli. A combined theoretical and experimental study has demonstrated a dynamical change in the electronic band structures and ∼60-fold photoluminescence enhancement upon cooling in MAPbBr_3_ hybrid perovskites. These were explained in terms of a decrease in the degree of rotational freedom of MA due to a change in the H-bonding interactions. H-bonding can also influence the mechanical properties of perovskite systems [67]. A recent study has reported on two Mn-based metal-organic framework perovskites [C(NH_2_)_3_][Mn(HCOO)_3_] and [(CH_2_)_3_NH_2_][Mn(HCOO)_3_] with the former giving higher Young’s moduli and hardness due to the stronger hydrogen bonding between the framework hosts and the amine cations [68]. Additionally, in the 2D HOIP, the inorganic and organic layers are connected via H-bonds, which provide structural stability to the overall geometry. It has been shown for some Mn (II)-based hybrid perovskites that stronger H-bonding interactions lead to larger PLQY [15,38,55]. Finally, for the studied samples here, one also needs to take into consideration the presence of the hydrated non-emissive octahedral phase that can further reduce the PLQY and affect their excited state dynamics.

To obtain more insight into the emission behavior, we recorded the emission decays, to be shown and discussed in Section 2.2.2. We first examine the temperature effect on the emission spectrum of P2. Recent reports have shown that a change in the temperature induces a conversion from one Mn(II) configuration to another one [31,69,70,71]. This effect is reflected in the emission color that switches from green to orange/red when the temperature increases, and the reverse when the temperature decreases [31,69]. The thermal stability of Mn-based perovskites is well documented, and thermogravimetric analysis of (CH_6_N_3_)_2_MnCl_4_ and 1-(2-aminoethyl)piperazine MnBr, for example, has shown large stability up to 573 K [38,72]. Thus, we explored the effect of the temperature (from 30 to 125 °C) on the emission spectrum of P1 and P2. To begin with P2, the results show a clear gradual decrease in intensity with the increase in temperature that does not vanish at 125 °C (Figure 4). The relatively small observed thermal quenching in the studied temperature range suggests a relatively long Mn-Mn distance. We also observed a small broadening of the emission band (inset of Figure 4). These results can be interpreted in terms of the opening of a non-radiative process or by the conversion of the Mn(II) tetrahedral configuration in P2 to an octahedral one. However, we did not observe any orange/red emission, the signature of the Mn(II) octahedral environment, as recorded for P1 (Figure 2A). Thus, we suggest that the decrease in the emission intensity of P2 upon increasing the temperature is rather due to an increase in the non-radiative processes, which are enhanced when electron-phonon interactions increase [31,71]. Notice that we did not observe any change in the excitation spectra upon increasing the temperature by almost 100 °C, which indicates that the emitters of P2 are originating from the same ground state population (Appendix A). We also calculated the activation energy barrier for the non-radiative processes for P2, Δ*E_a_*, following Equation (1) [38]:(1)$$I\left(T\right)=\frac{I_{0}}{1+\mathrm{Aexp}\left(-\frac{\Delta E_{a}}{k_{B}T}\right)}$$
where *I*(*T*) and *I*_0_ are the fluorescence intensities at temperatures *T* and *T*_0_ (303 K, 30 °C), respectively, A is the pre-exponential factor, Δ*E_a_* is the activation energy barrier of non-radiative processes and *k*_B_ is the Boltzmann constant. The fit to the temperature dependence of the emission intensity maximum gives Δ*E_a_* = 4.46 kJ/mol (Appendix A). For P1, Δ*E_a_* = 1.67 kJ/mol, which is about 3–4 times lower than that for P2, in agreement with the trend in the PLQY values of these perovskites (Appendix A). Note also that the thermal energy at room temperature (2.45 kJ/mol at 298 K) is slightly larger than Δ*E_a_* for P1 in agreement with its low PLQY. These energy barriers are lower than the one reported for (CH_6_N_3_)_2_MnCl_4_ perovskite (11.74 kJ/mol), having a PLQY of 55.9% [38].

#### 2.2.2. Time-Resolved Emission Data

To explore the photodynamical properties of the two Mn-based perovskites, we collected their emission decays in a PMMA matrix at several emission wavelengths upon excitation at 371 and 433 nm. No notable excitation and emission wavelength dependence was observed for both samples (Appendix A and Appendix A). The emission decays of both perovskites in the PMMA matrix decay biexponentially with time constants of τ_1_ = 83 µs (53%) and τ_2_ = 250 µs (47%) for P1, and τ_1_ = 100 µs (9%) and τ_2_ = 370 µs (91%) for P2 (Figure 5 and Table 1). These values give average lifetimes of τ_AVE_ = 212 µs and τ_AVE_ = 362 µs for P1 and P2, respectively. The larger value of τ_AVE_ for P2 agrees with its higher PLQY (26%) when compared to P1 (3.6%).

We assign the longer decay time (τ_2_) to the emission of not interacting Mn^2+^ ions, which eliminates the direct spin−spin interaction. On the other hand, the fastest component most likely originates from interacting Mn-Mn pairs. Reported PL lifetimes of the Mn-emission in hybrid materials comparable in structure to the ones in this study vary from a few nanoseconds to milliseconds depending on the structure, type of halide ions, the local environment of Mn^2+^, and most importantly, the Mn−Mn distance [28,56]. For example, in a study of [(CH_3_)_4_N]_2_MnBr_4_ and [(CH_3_)_4_N]MnBr_3,_ it was reported that the PL lifetime of the Mn^2+^ emission decreases by two orders of magnitude when the structure changes from the tetrahedral to the octahedral coordination [28]. The observed decrease in the PL decay lifetime was explained in terms of the relatively shorter Mn−Mn distance of 3.25 Å in [(CH_3_)_4_N]MnBr_3_ compared to 7.89 Å in [(CH3)_4_N]_2_MnBr_4_. Hence, we explain the decrease in τ_AVE_ when the perovskite structure changes from the tetrahedral coordination in P2 (τ_AVE_ = 362 µs) to the octahedral one in P1 (τ_AVE_ = 212 µs) with the decrease in the Mn-Mn distance in the latter. This will facilitate the dissipation of the excitation energy resulting in the increase of non-radiative recombination. Notice the relatively small difference between the values of the fluorescence lifetimes (τ_2_ is only 3–4 times longer than τ_1_) of the same perovskite, contrary to previous reports where larger changes were observed [28,56]. Thus, in the present perovskites, the difference in the Mn-Mn distances between the different emitting centers should not be very large. On the other hand, the possible presence of the non-emissive hydrated phase might also be the origin of the shorter time component with a value of 83 µs, shortening the value of τ_AVE_ for both samples.

Finally, to evaluate the effect of hydration on the photodynamics of the studied perovskites, we collected the emission decays of the unprotected, neat samples following excitation at 371 nm and gating at 600 nm for P1 and at 550 nm for P2. The observed decays show a notable increase in the contribution of the fast component (83–100 µs), which is more pronounced for P1 (from 53% in the PMMA matrix to 62% when unprotected, τ_AVE_ = 192 µs) than for P2 (from 9% to 14%, τ_AVE_ = 358 µs). These results agree with the observed higher sensitivity of P1 to the ambient humidity and with the formation of a hydrated, non-emissive quasi-octahedral phase that efficiently quenches the emission of both perovskites (Figure 3).

## 3. Materials and Methods

### 3.1. Materials

Manganese (II) bromide tetrahydrate (MnBr_2_·4H_2_O, 98%) was obtained from Acros Organics (Madrid, Spain). Poly(methyl methacrylate) (PMMA, Mw = 996 000 g.mol^−1^), ethylenediamine (EDA) (98%), and toluene were purchased from Sigma Aldrich (Madrid, Spain). Hydrobromic acid (HBr, 48% *w*/*w* aq. soln.) was obtained from Alfa Aesar (Madrid, Spain). The ethylamine (EA) solution (66–72% aq. soln.) was purchased from Sigma Aldrich (Madrid, Spain). All chemicals were used without further purification.

Preparation of ethylenediamine dihydrobromide (C_2_H_8_N_2_·2HBr, EDA(HBr)_2_): First, EDA (0.7 mL, 0.25 M) was dissolved in 20 mL of ethyl acetate in a round bottom flask. Next, 5 mL of HBr (48% in water) was dropwise added to the previous solution under a constant stirring at 0–2 °C in an ice-water bath for two hours. Finally, the obtained EDA(HBr)_2_ salt was washed several times using ethyl acetate, and the solvent evaporated in a rotatory evaporator at 40 °C.

Synthesis of ethyl ammonium bromide (CH_3_CH_2_NH_3_Br, EA(HBr)): EA (0.5 mL, 0.25 M) was first dissolved in 20 mL of ethyl acetate in a round bottom flask and then 2.5 mL hydrobromic acid was added dropwise in a stirring condition at 0–2 °C for two hours. The transparent homogeneous CH_3_CH_2_NH_3_Br solution was then evaporated at 60 °C in a rotary evaporator. A slight yellowish solid was obtained. The solid was then washed with ethyl acetate and dried in a rotary evaporator at 40 °C to obtain a pure crystalline solid of CH_3_CH_2_NH_3_Br. The washing was repeated several times to ensure the removal of excess HBr.

Synthesis of ethyl ammonium manganese (II) bromide (C_2_H_5_NH_3_MnBr_3_, P1): For the synthesis of ethyl ammonium manganese (II) bromide, 0.14 g of ethyl ammonium bromide (0.2 M) and 0.28 g of manganese (II) bromide tetrahydrate (0.2 M) were dissolved in 5 mL methanol at room temperature [15]. The solution was then evaporated at 40 °C for several days until a red emissive solid was obtained.

Preparation of ethylenediammonium manganese (II) bromide (C_2_H_4_(NH_3_)_2_MnBr_4_, P2): The synthesis procedure was a slight modification of the already reported one [16]. Ethylenediamine dihydrobromide (2 mmol, 0.22 g) and MnBr_2_·4H_2_O (2 mmol, 0.28 g) were dissolved in 2.5 mL 48% HBr. The resulting solution was heated to 80 °C for 3 h. Finally, the solvent was slowly evaporated at 60 °C for two days. The obtained crystals were stored in a vial without further drying (P2(H1)) and dried at 120 °C in the oven for 12 h (P2). To obtain the more hydrated P2 (P2(H2)), the sample was exposed to ambient humidity for 12 h.

### 3.2. Methods

Powder X-ray diffraction (PXRD): The X-ray diffraction patterns of polycrystalline perovskite powders were obtained using a PANalytical diffractometer (X’Pert Pro model) and an X Bruker D8 Advance. The conditions used were 45 kV, 40 mA, CuKα radiation, and a system of slits (soller-mask-divergence-antiscatter) of 0.04 rad-10 mm-1/8°-1/4° with an X’celerator detector.

Electron-spin paramagnetic resonance (EPR): EPR spectra were carried out at room temperature using a Bruker ELEXSYS E500 spectrometer operating at the X-band. The spectrometer was equipped with a super-high-Q resonator ER-4123-SHQ, the magnetic field was calibrated by a NMR probe and the frequency inside the cavity (~9.36 GHz) was determined with an integrated MW-frequency counter.

Steady-state UV−visible spectroscopy: The steady-state UV−visible diffuse reflectance spectra were recorded using a JASCO V-670 spectrophotometer equipped with a 60 mm integrating sphere ISN-723. The obtained signals were converted to Kubelka−Munk function F(R) = ((1 − R)^2^/2R), where R is the diffuse reflectance intensity from the sample. The emission spectra of the perovskites were recorded using a FluoroMax-4 (Jobin-Yvon) spectrofluorometer. This system incorporates an integrating sphere setup Quanta from Horiba coupled to the spectrofluorometer allowing for measurement of the photoluminescence quantum yield (PLQY).

Time-resolved emission: Picosecond (ps) emission decays were collected using a time-correlated single-photon counting (TCSPC) system. The samples were excited using 40 ps-pulsed (<1 mW, 40 MHz repetition rate) diode lasers (PicoQuant, Germany) centered at 371 and 433 nm. The system was equipped with a laser driver (PDL820B, PicoQuant, Germany) for burst operation that allows the detection of luminescence signals at time windows of several seconds. The emission decays were collected and analyzed through a TCSPC and multi-channel scaling board (TimeHarp260 (nano), PicoQuant, Berlin, Germany) The fluorescence signal was gated at a magic angle (54.7°) and monitored at 90° with respect to the excitation beam at discrete emission wavelengths. The experimental decays were analyzed using a multiexponential function and looking for the best residual distribution.

## 4. Conclusions

In this work, we reported on the photobehavior of two Mn (II) bromide perovskites using monovalent (C_2_H_5_NH_3_MnBr_3_, P1) and a bivalent (C_2_H_4_(NH_3_)_2_MnBr_4_, P2) alkyl interlayer spacers. The EPR experiments suggest that the coordination of the Mn(II) center in P1 is octahedral while in P2, it is tetrahedral. The emission of P1 is orange-red and does not have a large PLQY (3.6%), while that of P2 is green and has a larger PLQY (26%). The difference in their photobehavior relies on the distinct Mn(II) coordination, which reflects different couplings in electron-phonon and Mn-Mn interactions. The stability against moisture at ambient conditions largely increases upon encapsulation of both perovskites by a PMMA film. The emission decay does not depend on the observation wavelength or the excitation one. It shows two components in the microsecond regime—the shortest lifetime for the hydrated phases and the longest lifetime for the non-hydrated phases. From the temperature-dependent of P2 emission intensity, we found an activation energy barrier, Δ*E_a_* = 4.46 kJ/mol, for the non-radiative processes, most probably due to electron-phonon interactions. To the best of our knowledge, this is the first report comparing the effect of mono and bivalent spacer cation bounded to the Mn centers in Mn-based perovskites, using the same alkyl chain. Although in the present study, despite our best efforts, we were unable to obtain single crystals of P1 and P2, the findings presented here provide insights for a better structural design of Mn(II)-based perovskites, to increase their lighting performance and stability. Future efforts should be aimed at resolving the crystal structure of perovskites with organic ligands with linear chains, as the ones described here, while theoretical calculations supported by the resolved crystal structure should provide details about band structure dynamics [63]. Furthermore, successfully controlling the long-term stability of 2D HOIP advanced time-resolved experimental techniques could resolve the ultrafast dynamics in these systems, which has been identified as one of the five fundamental challenges of 2D perovskite materials [44,59].

## Data Availability

The data are available from the authors upon request.

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
