# Peer review of "The Effects of Mono- and Bivalent Linear Alkyl Interlayer Spacers on the Photobehavior of Mn(II)-Based Perovskites"

_ijms, 2023, doi:10.3390/ijms24043280_

Round 1
Reviewer 1 Report
Authors of this work have reported on the properties of two new Mn (II) bromide perovskites using monovalent (C2H5NH3MnBr3, P1) and a bivalent (C2H4(NH3)2MnBr4, P2) alkyl 420 interlayer spacer. While this work is purely experimental based, it lacks several features. Some of them are summarized below:
1- The geometrical aspects of the perovskite systems are not analyzed with care. What kind of bonding features existing between the organic and inorganic moieties that provide stability to the overall geometry of the systems analyzed?
2- I believe hydrogen bonding, and other interactions, should be playing an important role in the systems reported. I cannot see any discussion of this in the ms. There are good amount of research articles reported focusing hydrogen bonding in halide perovskite semiconductors, but such studies are not cited.
3- There are some grammatical errors scattered throughout the ms.
4- I cannot see any DFT calculations in this study. The bandgap features of the compounds are not clearer.
5- Does Scheme 1 represent the “unit-cells” of the crystals analyzed? I cannot see the CIF files as supplementary information. Surprisingly, the Lattice properties are not discussed.
Author Response
Reviewer 1.
Response to a general comment: We would like to point out that due to the high sensitivity to water and characteristics of the used compounds, we could not obtain single crystals of the perovskites even after many attempts and different ways. Thus, the results for the structural characterization part in the submitted manuscript rely on powder X-ray and EPR data.
Comments
- The geometrical aspects of the perovskite systems are not analyzed with care. What kind of bonding features existing between the organic and inorganic moieties that provide stability to the overall geometry of the systems analyzed?
The perovskites reported here are ethylammonium manganese (II) bromide (C2H5NH3MnBr3, P1) and ethylenediammonium manganese (II) bromide (C2H4(NH3)2MnBr4, P2). P1 shows red emission whereas P2 gives green emission. The geometry and coordination have been proposed as per the emission maxima. It is well established for Mn (II) based perovskites that green emission always comes from the tetrahedral arrangements of the Mn (II) and red emission manifests octahedral arrangement. In addition to that, our EPR data fully agree with the suggested Mn(II) configuration in P1 and P2. Unfortunately, even after many attempts, we could not get single crystals of the reported perovskites thus, we are unable to point out the precise bonding interactions.
We have added the following text in the manuscript (Page 7, lines 290 - 300):
“Additionally, the organic-inorganic metal halide hybrid perovskites are generally considered as van der Waals hybrid systems and formed by stacking inorganic metal halide layers confined between two organic layers.[35,39,57] The inorganic and organic layers are connected via H-bonding interactions, which provide structural stability to the overall geometry. It has been shown for some Mn (II)-based hybrid perovskites that stronger H-bonding interactions lead to larger PLQY.[15, 39, 57] In P1 and P2, the organic cations have linear chains that render higher flexibility in the resulting lattice, which would enhance the electron-photon interactions and might in part explain the relatively lower PLQY values. Furthermore, we suggest that for P2 the H-bonding between the [MnBr4]2- unit and the EDA cation is much stronger than that of the [MnBr6]4- and the EA cation” (Page 7, lines 290 - 300, Ref. 15, 35, 39 and 57)
We have also added a suggestion in the caption of Scheme 1.
- I believe hydrogen bonding, and other interactions, should be playing an important role in the systems reported. I cannot see any discussion of this in the ms. There are good amount of research articles reported focusing hydrogen bonding in halide perovskite semiconductors, but such studies are not cited.
Yes, we agree with the Referees, and we have included a short discussion and relevant references regarding the H-bonding interactions (See reply to comment 1). (Page 7, lines 290 – 300, Ref. 15, 35, 39 and 57). We have also included the expected H-bonding interaction in Scheme 1 for convenience and modified the Scheme 1 caption (page 3, lines 128 and 129).
- There are some grammatical errors scattered throughout the ms.
We have thoroughly checked the manuscript and the grammatical errors have been eliminated in the revised version. Hope all is now fixed.
- I cannot see any DFT calculations in this study. The bandgap features of the compounds are not clearer.
Since we could not obtain single crystals of the perovskites, we could not perform DFT calculations to theoretically get the band structure features. In the Introduction, we have added a text highlighting finding of the theoretical calculations to predict few properties of the perovskite systems, and added few references (page 2, lines 93 – 98, Refs. 16, 17, 39, 41-43):
“To gain insight into the electronic structure, origin of bright photoluminescence and ferromagnetic coupling in this kind of materials, several ab initio models have been proposed.[42, 43] Recently, these methods have been applied to predict the band structure and to validate the origin of Mn2+ photoluminescence through spin forbidden 4T1 → 6A1 transitions for some all-Mn halide perovskite single crystals.[16, 17, 39, 41]”
- Does Scheme 1 represent the “unit-cells” of the crystals analyzed? I cannot see the CIF files as supplementary information. Surprisingly, the Lattice properties are not discussed.
As we said previously, unfortunately we could not obtain single crystals for these perovskites. Thus, we cannot provide CIF files and related discussion on the lattice properties. Scheme 1 does not represent “unit-cells”. It represents a proposed Mn2+ coordination in the two perovskites. Our proposal is based on the recorded emission and EPR spectra of these perovskites. Please note that it is well established that Mn2+ tetrahedral coordination results in a green emission, while a an octahedral coordination gives a red one. We wrote about this in the Introduction and in the Results and Discission sections.

Reviewer 2 Report
This manuscript reports on the effect of mono- and bivalent alkyl interlayer spacer on the photo-behavior of Mn(II)-based bromide perovskites. The research results indicate that the emission intensity of both of the two perovskites considered shows a large sensitivity to moisture (air humidity, 50-60 %) and a significantly improved stability upon encapsulation within a poly(methyl methacrylate) (PMMA) film. The novel results consisting in comparing the effect of mono and bivalent spacer cation bounded to the Mn centers in Mn-based perovskites (employing the same carbon chain) are very well transmitted by the discussion in the present manuscript.
The research questions as defined in this work are very much worth investigation while in the present context the specific perovskites material and demonstrator systems, and the evaluation of the effect of mono- and bivalent alkyl interlayer spacers is very well analyzed, specifically explained and comprehensively narrated. The characterization efforts (including PXRD and the photophysical characterization) as employed are not only adequate for the present purpose, but also the interpretation of characterization results seems correctly done.
All in all, the results are interesting and presented in a way which is easy and valuable for a wider audience of readers. All these results seem like worthy points of departure for the discussion and are plausible and well substantiated in the view of the conclusions drawn.
The reported results bring new knowledge and certainly represent an original contribution with probable technological impact in novel sensor materials.
The authors chose an adequate structure of the manuscript for such a study. Also, they provided a balanced realistic and nicely illustrated presentation of their results and corresponding analysis that is of much scientific and practical interest and adds new knowledge to the field.
The present manuscript is a significant contribution, this work once published would be instructive and suggestive in terms of further studies and with good chances be cited.
There are some minor issues with this already excellent manuscript that will need to be addressed before the manuscript becoming suitable for publication, i.e., it can be considered for publication after a minor revision:
1: Authors are a little bit too telegraphic in what concerns the characterization methods (which are much adequate) especially in the abstract (where short mentioning of characterization techniques is missing). It will publicize their work better if they are more detailed/descriptive concerning this issue.
2: Title is not optimal too. The word “behavior” is usually used in singular only (especially in the present context); “two new” is obsolete too, number is not so important to mention in the title while “new” is a banality and thus is obsolete and must be omitted. Writing all substantives and adjectives with capital letters is, in principle, incorrect in English and should be avoided (except if the journal explicitly requires such a practice).
3: Are the authors aware of any studies (including theoretical) of the thermal stability (maximum temperature at which the compound is preserved) for the Mn(II)-based bromide perovskites? If so, this should be reflected in the manuscript.
4: In the introduction, the authors miss that structural features/interatomic distances/ coordination of compounds with similar structural complexity to the Mn(II)-based bromide perovskites, can be understood in relation to their synthesis by employing specially developed theoretical and ab initio methodology [e.g., Dalton Transactions 44 (2015) 3356-3366, and Carbon 81 (2015) 620-628] with direct practical implications for the credibility of the claims about structural/coordination findings. This aspect should be acknowledged in the present manuscript.
5: Spell-check and stylistic revision of the paper are necessary. Some long sentences, as well as misspellings, etc., are noticeable throughout the text.
Author Response
Please see attached
Reviewer 2.
Recommendation: Minor revisions needed as noted.
- Authors are a little bit too telegraphic in what concerns the characterization methods (which are much adequate) especially in the abstract (where short mentioning of characterization techniques is missing). It will publicize their work better if they are more detailed/descriptive concerning this issue.
We thank the Reviewer for pointing this out. We have modified the abstract and included the characterization techniques used in this work (Page 1, lines 14 - 18):
̏The perovskites were characterized with powder X-ray diffraction (PXRD), electron spin para-magnetic resonance (EPR), steady-state and time-resolved emission spectroscopy. The EPR experiments show the signatures of octahedral coordination in P1 and tetrahedral one for P2, while the PXRD results demonstrate presence of hydrated phase in P2 when exposed to ambient conditions.˝
- Title is not optimal too. The word “behavior” is usually used in singular only (especially in the present context); “two new” is obsolete too, number is not so important to mention in the title while “new” is a banality and thus is obsolete and must be omitted. Writing all substantives and adjectives with capital letters is, in principle, incorrect in English and should be avoided (except if the journal explicitly requires such a practice).
We have modified the title following the suggestion of the Referee. The new title is now: ̏ The Effect of Mono- and Bivalent Linear Alkyl Interlayer Spacer on the Photobehavior of Mn(II)-based Perovskites˝.
- Are the authors aware of any studies (including theoretical) of the thermal stability (maximum temperature at which the compound is preserved) for the Mn(II)-based bromide perovskites? If so, this should be reflected in the manuscript.
The thermal stability of Mn-based bromide perovskites has been reported in several studies. We have clarified this point in the manuscript following the reviewer´s suggestion. (Page 7, lines 308-310, Refs. 39 and 61). For the present perovskites, we have not done any study on the thermal stability. We will consider this point in further studies.
“The thermal stability of Mn-perovskites is well documented, and thermogravimetric analysis of (CH6N3)2MnCl4 and 1-(2-aminoethyl)piperazine MnBr, for example, has shown large stability up to 573K.[39, 61].”
- In the introduction, the authors miss that structural features/interatomic distances/ coordination of compounds with similar structural complexity to the Mn(II)-based bromide perovskites, can be understood in relation to their synthesis by employing specially developed theoretical and ab initio methodology [e.g., Dalton Transactions 44 (2015) 3356-3366, and Carbon 81 (2015) 620-628] with direct practical implications for the credibility of the claims about structural/coordination findings. This aspect should be acknowledged in the present manuscript.
We thank the Reviewer for the suggestion. We have now included new references and modified the Introduction. We believe that the suggested references, although of very high quality, fall outside the scope of the current work. Thus, we have included/cite works that are more centered on the Mn2+ properties in these complex environments (perovskites) (page 2, lines 93 – 98, Refs. 16, 17, 39, 41-43). We hope that the Reviewer understands our point of view.
“To gain insight into the electronic structure, origin of bright photoluminescence and ferromagnetic coupling in this kind of materials, several ab initio models have been proposed.[42, 43] Recently, these methods have been applied to predict the band structure and to validate the origin of Mn2+ photoluminescence through spin forbidden 4T1 → 6A1 transitions for some all-Mn halide perovskite single crystals.[16, 17, 39, 41]”
- Spell-check and stylistic revision of the paper are necessary. Some long sentences, as well as misspellings, etc., are noticeable throughout the text.
We have thoroughly checked the manuscript and eliminated the grammatical errors. We believe that all is OK now.

Round 2
Reviewer 1 Report
1- First of all, I cannot see any revision in the ms since the changes made are NOT highlighted. I am not sure what part of the paper is changed!! I am also not satisfied with the reply provided by the authors. In all occasions, their reply is negative in the sense they were unable to have a crystals of the systems that were the focus of the study. If it is the case, then where is point of stability and how would the study assist other researchers to envision the impact of the perovskite system? And how would the study resolve the long-standing issue of innovating environmentally friendly (and stable) metal halide perovskites for social needs?
2- Authors wrote that “Unfortunately, even after many attempts, we could not get single crystals of the reported perovskites thus, we are unable to point out the precise bonding interactions.” If it is the case, then how come a researcher of the field will rely on the speculations made of this paper?
3- It is purely misleading to say that “Additionally, the organic-inorganic metal halide hybrid perovskites are generally considered as van der Waals hybrid”. There is a clear distinction between van der Waals interactions in chemical systems, and those are driven by hydrogen bonding. See the following papers (J. Phys. Chem. Lett. 2017, 8, 24, 6154–6159; Int. J. Mol. Sci. 2022, 23(15), 8816; Chem. Mater. 2017, 29, 14, 5974–5981; Scientific Reports 6, Article number: 21687 (2016); Scientific Reports volume 9, Article number: 50 (2019); https://www.sciencedirect.com/science/article/pii/S2468519418300338; ), for examples, which are not even referred. Accordingly, authors should modify their claims by briefing and citing the literature just mentioned above.
I suggest the authors to write the weakness of the study so the readers of the work can understand the underlying issues involved, and also provide directions how to resolve them.
Author Response
February 1st, 2023
Subject: Revised manuscript (Manuscript ID: ijms-2191169)
Dear Editor,
Thank you for considering our manuscript, “The Effect of Mono- and Bivalent Linear Alkyl Interlayer Spacer on the Photobehavior of Mn(II)-based Perovskites” for publication in International Journal of Molecular Sciences. Below we provide our response (in blue) to the comments and suggestions of Reviewer 1. In this revised version, we have addressed the comments and suggestions in a positive way by providing further clarifications to substantiate our results. Changes done to the main text are highlighted in yellow in the MARKED revised version of the manuscript. In green we have highlighted changes made in the previous round of revision. We would like to thank the Referee for the dedicated time and for the valuable and constructive comments.
Reviewer 1:
1- First of all, I cannot see any revision in the ms since the changes made are NOT highlighted. I am not sure what part of the paper is changed!!
We apologize for the confusion with submitting a file without highlighting the changes. Unfortunately, the unmarked file was saved under the incorrect name and was submitted to the system as the highlighted one. This is why we always also provide a detailed description of the changes in the response letter. In the newly submitted version we have highlighted the previous changes in green and the newly introduced ones in yellow.
- I am also not satisfied with the reply provided by the authors. In all occasions, their reply is negative in the sense they were unable to have a crystals of the systems that were the focus of the study. If it is the case, then where is point of stability and how would the study assist other researchers to envision the impact of the perovskite system? And how would the study resolve the long-standing issue of innovating environmentally friendly (and stable) metal halide perovskites for social needs?
We believe that there might have been some misunderstanding as to what the goals of our study are. It is true that obtaining single crystal structure would have provided better understanding of the crystal structure of the two perovskites. In page 4, lines 148 – 165 and page 5, Lines 205 – 213 (added Refs: 15, 39, 44, 45 and 56), we added texts where we explain the difficulty in obtaining single crystals of P1 and P2 and discuss the possible origin for this difficulty. Note that this is not the primary goal of this work. The reported characterization and spectroscopic studies do not depend on reporting single crystal data. The reached conclusions rely mostly on the observed behavior from the reported experiments. When discussing our results, we have made a comparison with published works. Furthermore, even if single crystals were obtained, we don´t expect that this would change the overall photophysical view on the studied systems, neither would give information on the perovskite stability. About the stability of these perovskites, we have indicated that they suffer from humidity at ambient conditions, but they are robust when PMMA-protected from moistures. The information is in: Page 1, Line 22; Page 3, Line 111 and Figure 3.
With respect to the “long-standing issue of innovating environmentally friendly (and stable) metal halide perovskites for social needs”, there is obviously a long way between studies like this (and many others published in this field) and resolving the global issues of perovskite stability and efficiency for social needs. However, we believe that studies like the present one can help in achieving this goal.
3- Authors wrote that “Unfortunately, even after many attempts, we could not get single crystals of the reported perovskites thus, we are unable to point out the precise bonding interactions.” If it is the case, then how come a researcher of the field will rely on the speculations made of this paper?
We share the Referee´s disappointment regarding the lack of single crystal data. However, although we could not obtain this data, the results from the EPR, steady-state and time-resolved experiments form the base for our suggestions, discussion and conclusions. These are further compared to published studies on similar systems, which provide additional support. In general, we try to avoid unsubstantiated and unnecessary speculations, and when it is unavoidable, it is clearly indicated in the text.
As per Referee´s suggestion, we have modified the Conclusions part to address the challenges encountered in this work (Page 12, Lines 506 – 515). We have added two new reference (Refs. 45 and 60)
- It is purely misleading to say that “Additionally, the organic-inorganic metal halide hybrid perovskites are generally considered as van der Waals hybrid”. There is a clear distinction between van der Waals interactions in chemical systems, and those are driven by hydrogen bonding. See the following papers (J. Phys. Chem. Lett.2017, 8, 24, 6154–6159; Int. J. Mol. Sci. 2022, 23(15), 8816; Chem. Mater. 2017, 29, 14, 5974–5981; Scientific Reports volume 6, Article number: 21687 (2016); Scientific Reports volume 9, Article number: 50 (2019); https://www.sciencedirect.com/science/article/pii/S2468519418300338; ), for examples, which are not even referred. Accordingly, authors should modify their claims by briefing and citing the literature just mentioned above.
We thank the Referee for pointing out this confusion. We have modified the text following the Referee´s suggestions (pages 7 - 8, lines 292 - 358). Since the modified text is too long, we have not copied the modified part in the response letter.
We have also included the suggested references (5 references) and few more (4 references) in the text (except for “Scientific Reports volume 9, Article number: 50 (2019)”, which studies systems (Bi2Se3 nanoribbons) and processes not related to the ones discussed in the present work). Refs: 59 - 62, 64 – 69.
- I suggest the authors to write the weakness of the study so the readers of the work can understand the underlying issues involved, and also provide directions how to resolve them.
We have followed the suggestions of the Referee and included an explanatory texts at relevant places in the main ms: a) in the structural characterization part where we explain the difficulty in obtaining single crystals of P1 and P2 and discuss the possible origin for this difficulty (Page 4, lines 148 – 165 and Page 5, Lines 205 - 213, Refs: 15, 39, 44, 45 and 56), and b) in the Conclusions part where we provide a perspective as to what the next steps for these systems should be (Page 12, Lines 531 – 540, Refs: 45 and 60).
We hope that we have addressed satisfactory the Referee´s comments, suggestions, and critiques, we have added 12 new references and have improved our manuscript for publication in the IJMS.
Kind regards;
Prof. Abderrazzak Douhal

Round 3
Reviewer 1 Report
Authors have considered my comments and revised their paper. I am OK with the revisions made, even though there are things that the authors did not address properly (calling it as a misunderstanding and/or dissapointment!). I suggest acceptance of this work for possible publication